# SIEVE: MULTIMODAL DATASET PRUNING USING IMAGE CAPTIONING MODELS

## ABSTRACT

Vision-Language Models (VLMs) are pretrained on large, diverse, and noisy web-crawled datasets. This underscores the critical need for dataset pruning, as the quality of these datasets is strongly correlated with the performance of VLMs on downstream tasks. Using CLIPScore from a pretrained model to only train models using highly-aligned samples is one of the most successful methods for pruning. We argue that this approach suffers from multiple limitations including: 1) false positives due to spurious correlations captured by the pretrained CLIP model, 2) false negatives due to poor discrimination between hard and bad samples, and 3) biased ranking towards samples similar to the pretrained CLIP dataset. We propose a pruning method, SIEVE, that employs synthetic captions generated by image-captioning models pretrained on small, diverse, and well-aligned image-text pairs to evaluate the alignment of noisy image-text pairs. To bridge the gap between the limited diversity of generated captions and the high diversity of alternative text (alt-text), we estimate the semantic textual similarity in the embedding space of a language model pretrained on billions of sentences. Using DataComp, a multimodal dataset filtering benchmark, we achieve state-of-the-art performance on the *large* scale pool, and competitive results on the *medium* scale pool, surpassing CLIPScore-based filtering by 1.7% and 2.6% on average, on 38 downstream tasks.

## 1 INTRODUCTION

CLIP (Contrastive Language-Image Pre-training) (Radford et al., 2021) models have shown great success in solving zero-shot image classification and multimodal retrieval tasks. In addition, many foundational Vision-Language Models (VLMs) use pretrained CLIP encoders to condition image generation on CLIP text embeddings (Ramesh et al., 2022) in retrieval augmented vision-language models (Hu et al., 2023; Yasunaga et al., 2023), and to align modalities including audio, depth, and thermal with language through CLIP image embeddings (Girdhar et al., 2023). Therefore, the quality of CLIP representations can influence the performance of many VLMs.

To pretrain CLIP, billions of image-text pairs are collected using common crawl. The raw data is highly diverse but contains many noisy image-text pairs, including low quality images, low quality alternative text (alt-text), and misaligned image-text pairs. Pretraining CLIP models on noisy data can have adverse effects on the learned representations, thus leading to poor performance on downstream tasks (Abbas et al., 2023).

To address this challenge, researchers have developed data pruning methods to remove low quality image-text pairs. Heuristics that filter out image-text pairs based on image dimensions, aspect ratio, alt-text length, and complexity are commonly used (Schuhmann et al., 2021; Gadre et al., 2023) to reduce noise, but can also limit the diversity of the dataset (Nguyen et al., 2023). Methods that use images or class names from datasets, like ImageNet, to sample semantically similar image-text pairs can lead to higher accuracy on downstream tasks (Xu et al., 2023), but limit the diversity of the selected samples as they sample image-text pairs close to a specific dataset.

One of the most effective pruning methods, CLIPScore (Hessel et al., 2021; Schuhmann et al., 2021), computes the cosine similarity between image and text embeddings using a pretrained CLIP model. This score is then used to rank the alignment of image-text pairs. However, as shown in Figure 1, using CLIPScore can lead to false positives – samples that are poorly aligned but have

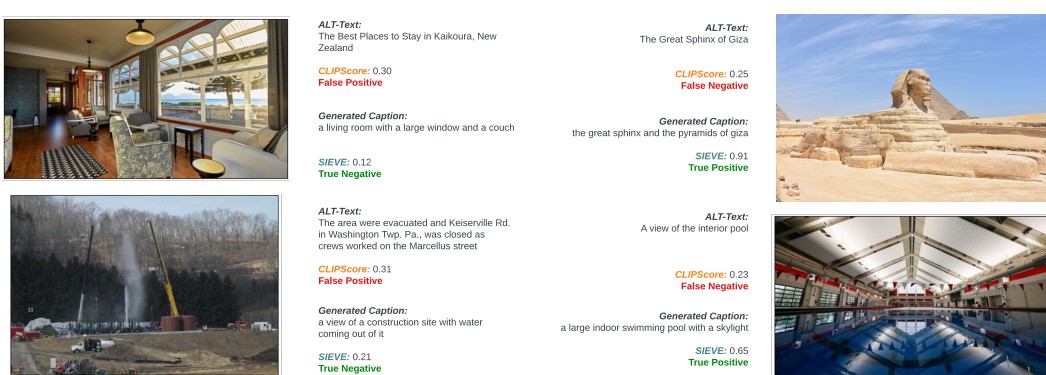

Figure 1: Examples of image-text pairs in which the scores of pretrained CLIP models, a commonly used image-text data filtering approach, fail to measure their alignment. Our proposed approach, SIEVE, provides an accurate alignment score using a caption generator and sentence transformer. **Top left** and **bottom left**: Examples of false positives where alt-text describes concepts that are not found or unrelated to the image. CLIP is trained on similar noisy image-text pairs, thus, it assigns a relatively high score. SIEVE can detect that such image-text pairs are misaligned. **Top right** and **bottom right**: Examples of false negatives where images are aligned with the alt-text but are assigned low CLIP scores, either due to the low likelihood of these pairs in the pretraining data, or because CLIP may have seen similar images aligned with other noisy labels. SIEVE can detect that such image-text pairs are well-aligned and selects them for pretraining.

high CLIPScore (i.e., bad samples) due to spurious correlations learned by the pretrained CLIP model (Yang et al., 2023). In addition, using CLIPScore can lead to false negatives – samples that are aligned but have low CLIPScore (i.e., hard samples) due to the poor discrimination between hard and bad samples. Excluding hard samples and including bad samples can negatively affect the generalization of CLIP image and text encoders.

The goal of this work is to reduce both false negatives and positives induced by CLIPScore ranking by relying on an image-captioning model pretrained on small, diverse, and well-aligned image-text pairs. As depicted in Figure 2, we evaluate the alignment of web-crawled image-text pairs by first, generating multiple captions for each image using nucleus sampling (Holtzman et al., 2020), followed by removing phrases that describe the medium (e.g., "an image of", "a photo of") rather than visual concepts. Finally, to evaluate semantic similarity between the limited diversity of generated captions and the high diversity of alt-text, we utilize the embedding space of a lightweight sentence transformer pretrained on billions of text pairs. The alignment between the generated captions and the alt-text is then used as a proxy for image-text alignment.

To evaluate the effectiveness of our proposed pruning method, we utilize the DataComp (Gadre et al., 2023) benchmark, which fixes the pretraining hyperparameters of CLIP and provides multiple candidate pools of noisy image-text data for pretraining CLIP models. The goal is to select a subset of noisy image-text data that leads to the best performance on 38 downstream tasks. Using image-captioning model alignment scores fused with CLIPScore, we achieve state-of-the-art performance on the *large* scale and competitive results on the *medium* scale, surpassing CLIPScore-based filtering by 1.7% and 2.6%, respectively, on average, on all 38 downstream tasks.

## 2 RELATED WORK

**Heuristics** are basic filtering methods including: filtering non-English alt-text using fastText (Joulin et al., 2016), filtering alt-text with a few words (Schuhmann et al., 2021; Gadre et al., 2023), filtering alt-text with low text complexity (Radenovic et al., 2023), and filtering images by size or aspect ratio (Gadre et al., 2023). A combination of these unimodal filtering approaches has been explored by DataComp (Gadre et al., 2023). An example of a multimodal filtering approach is text spotting: detecting and recognizing text in images and filtering image-text pairs with high overlap between spotted text (text detected in image) and alt-text (associated label of image) (Radenovic et al., 2023).

**Datasets as Priors** was proposed in DataComp (Gadre et al., 2023), relying on sampling image-text pairs that are semantically similar to diverse and curated datasets like ImageNet (Deng et al.,

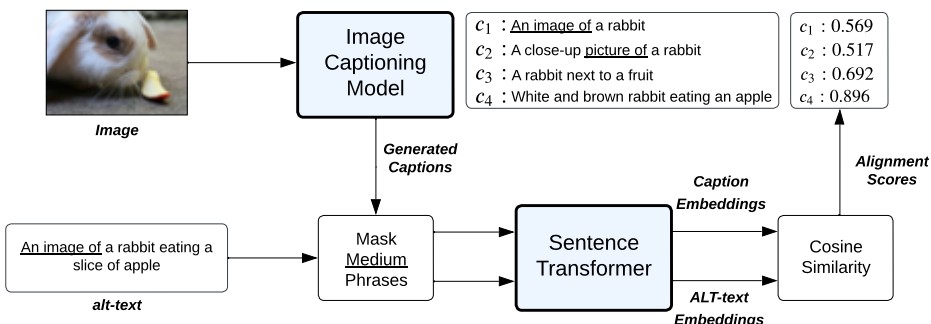

Figure 2: Our proposed framework enables dataset pruning using image-captioning models. To evaluate the alignment of a noisy image-text pair, we generate multiple captions per image using nucleus sampling. Then medium phrases, like "an image of" or "a photo of", are masked from alt-text and generated captions. Finally, a lightweight sentence encoder is used to semantically compare the generated captions with alt-text.

2009). Text-based sampling selects image-text pairs with alt-text overlapping one of the ImageNet classes. CiT (Xu et al., 2023) uses cosine similarity to filter alt-text that are similar to ImageNet classes. Image-based sampling approaches encode images from the unfiltered candidate pool using the OpenAI CLIP's ViT-L/14 vision encoder, and clusters the images into 100,000 groups using FAISS (Johnson et al., 2019). Then, embeddings of ImageNet training samples are used to keep the closest cluster to each sample. The main limitation of such approaches is that they bias the CLIP model and may not generalize well to new downstream tasks. Our approach, SIEVE, does not use any dataset as a prior.

**Pretrained VLMs** One of the most successful methods for evaluating image-text alignment is CLIPScore (Hessel et al., 2021). LAION filtering (Schuhmann et al., 2021) uses an OpenAI CLIP model (Radford et al., 2021) pretrained on 400 million image-text pairs to evaluate image-text alignment of large webscale datasets, and filter out samples with the lowest CLIPScore. Filtering using CLIPScore can suffer from false negatives, which leads to filtering out hard informative samples, and false positives, which leads to including misaligned samples. Inspired by text spotting (Radenovic et al., 2023), T-MARS (Maini et al., 2023) is concurrent work that detects and masks text regions in images before computing CLIPScore. Another concurrent work proposes a non-filtering approach that utilizes pretrained VLMs (Nguyen et al., 2023), using large image-captioning models like BLIP2 (Li et al., 2023) to replace alt-text labels with descriptive synthetic captions. The synthetic captions are then used to train CLIP models. The authors (Nguyen et al., 2023) demonstrate that at scale, the improvement of synthetic captions is capped by the limited diversity of generated captions compared to the high diversity of noisy text labels. Compared to (Nguyen et al., 2023), we do not alter the original alt-text and thus our focus is on the dataset pruning challenge.

## 3 METHODOLOGY

Let $\mathcal{D} = \{(I_i, T_i)\}_{i=1}^{N}$ denote an uncurated dataset consisting of $N$ image-text pairs crawled from the web. Our goal is to curate a dataset, $\mathcal{D}' = \{(I_{i'}, T_{i'})\}_{i'=1}^{N'}$, that is a subset of the pool, $\mathcal{D}' \subseteq \mathcal{D}$, $N' \leq N$, to train a new CLIP model from uninitialized weights, $\Theta_0$, to new weights, $\Theta'$:

$$\Theta' = \text{train}(\Theta_0, \mathcal{D}') \tag{1}$$

For a given scoring function, $f$, that maps an image-text pair to a scalar value, $s = f(I_i, T_i)$, we express a pruning function, $\mathcal{D}_f$, that selects a fraction, $k$, of dataset, $\mathcal{D}$, using function, $f$:

$$\text{prune}_f(\mathcal{D}, k) \qquad \text{s.t. } 0 \leq k \leq 1, \ f : (I_i, T_i) \to \mathbb{R} \tag{2}$$

where $\text{prune}_f(\mathcal{D}, k)$ applies function, $f$, on each image-text sample in $\mathcal{D}$ to obtain a score for each sample, ranks the scores in descending order, and returns a set of the top $k$ portion of the samples.

One common approach for pruning is CLIPScore (Gadre et al., 2023; Schuhmann et al., 2021). Let $E$ be a CLIP model consisting of an image encoder, $E_{\text{image}}$, that maps an image, $I$, to an embedding

vector, $E_{\text{image}}(I) \in \mathbb{R}^d$, and a text encoder, $E_{\text{text}}$, that maps a text sample, $T$, to an embedding vector, $E_{\text{text}}(T) \in \mathbb{R}^d$. CLIPScore is a measure of alignment between $I$ and $T$, and is defined as:

$$f_{\text{CLIP}}(I, T) = \langle E_{\text{image}}(I), E_{\text{text}}(T) \rangle \tag{3}$$

where $\langle \mathbf{x}, \mathbf{y} \rangle$ is the cosine similarity between two vectors, $\mathbf{x}$ and $\mathbf{y}$, which is defined as the dot product of the $l_2$ normalized vectors. The most common CLIP model used for pruning is pretrained on 400 million noisy image-text pairs Schuhmann et al. (2021). Our proposed pruning method, SIEVE attempts to minimize the false positives and negatives induced by CLIPScore filtering. SIEVE consists of two main components: Image-Captioning and Sentence Transformer.

**Image-Captioning** Let $G$ be a captioning model that generates text, $T_i^G$, describing the content of image, $I_i$:

$$T_i^G = G(I_i) \tag{4}$$

Given a captioning model pre-trained on a small, representative and well-aligned dataset of image-text pairs, we are interested in estimating the alignment between image-text pairs sampled from a very large, diverse but noisy dataset. The alignment score can then be used as a ranking metric for dataset filtering. We hypothesize that:

- The probability of generating a caption that is semantically similar to the alt-text from an aligned pair is much higher than that from a misaligned pair.

- The probability of generating a caption that is semantically similar to a hard alt-text is higher than generating a caption that is semantically similar to a misaligned alt-text. Here, a hard alt-text is a text label with low likelihood with respect to the captioning model, but is aligned with the image content.

As images can contain multiple objects with complex attributes and relationships, there exist multiple ways to describe their content. Given the inherent many-to-many relationship between images and text labels, our goal is to increase the probability of generating a caption that matches an aligned alt-text. To achieve this, we utilize nucleus sampling (Holtzman et al., 2020), a decoding strategy used to sample multiple captions, $r$, per image:

$$G(I, r) = \{T_0^G, T_1^G, \ldots, T_{r-1}^G\} \tag{5}$$

**Sentence Transformer** Given an image, its alt-text, and a set of generated captions, our goal is to estimate the alignment between the generated captions and the alt-text. However, there is a very large diversity gap between the generated captions and the highly diverse alt-text as measured by the number of unique nouns and trigrams (Nguyen et al., 2023). On the other hand, constructing a large, diverse and curated image-text dataset is expensive, which limits the diversity of the generated captions. We propose to bridge this gap by utilizing a light-weight sentence similarity model to encode the alt-text and the generated captions. We expect the semantically similar alt-text and generated caption pairs to be closely clustered in the embedding space compared to semantically distinct pairs. We reason that the rich semantic textual embedding space of the sentence similarity model enables pretraining the captioning model only on a small but curated image-text dataset. Thus, we rely on the semantic understanding of the sentence similarity model to bridge the gap between the limited diversity in the generated captions and the highly diverse alt-text labels.

To estimate the alignment score, we compute the cosine similarity between embeddings of each generated caption and text label. Let $S$ be a language model that encodes a text sample, $T$, to a vector, $S(T) \in \mathbb{R}^d$. We define the alignment between two text samples, $T_a$ and $T_b$, as the cosine similarity between their language model encodings:

$$\langle S(T_a), S(T_b) \rangle \tag{6}$$

This estimate can then be used as a proxy for the image-text alignment of an image, $I$, and text, $T$:

$$\langle S(G(I)), S(T) \rangle$$

If the image captioning model generates $M$ different caption candidates for an image, $I$, we can use the maximum alignment between each of the generated captions, $G(I, r) = \{T_0^G, T_1^G, \ldots, T_{r-1}^G\}$, and a text sample, $T$:

$$\max_{T_j^G \in G(I,r)} \langle S(T_j^G), S(T) \rangle \tag{7}$$

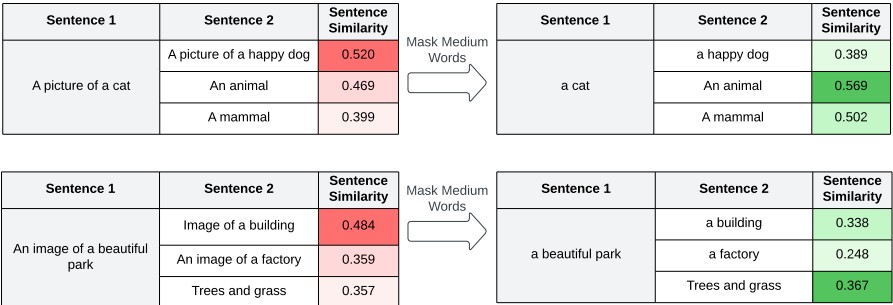

Figure 3: Masking medium phrases improves the ranking of sentence similarity scores. On the **left**, sentence pairs with misleadingly high (or low) sentence similarity due to the existence (or absence) of medium phrases are highlighted in dark red (or light red). On the **right**, similarity scores that are more aligned with semantics are highlighted in dark green. The sentence similarity scores are computed using the all-MiniLM-L6-v2 sentence transformer (Wang et al., 2020).

In literature, there are different models and approaches to obtain text embeddings. Joulin et al. (2017) use the average of N-gram features of each word in a text sample to obtain an embedding. A more common option is to use the logits of the last token generated by a decoder-only language model, which is the approach taken with CLIP's text encoder (Radford et al., 2021), in GPT 1 (Radford & Narasimhan, 2018), as well as in Abbas et al. (2023). Encoder-only models, such as BERT (Devlin et al., 2018) or RoBERTa (Liu et al., 2019), can also be used, where the embedding vector may be either the logits of the classification token, or the average pool of the logits of all tokens. Although such language models may have strong generation or classification capabilities, they were not optimized for sentence similarity tasks, but either for next word prediction (i.e., causal language modeling) or masked word prediction (i.e., masked language modeling) tasks. Therefore, their embeddings may not be ideal to measure alignment between sentences. More importantly, such models are large in size and hence slow to infer on large datasets. A language model finetuned on a sentence similarity task, such as SNLI (Bowman et al., 2015), aligns with the goal of estimating semantic textual similarity between alt-text and captions. We find that sentence similarity models (Reimers & Gurevych, 2019) pretrained using a self-supervised instance discrimination task on billions of sentences perform well in estimating the alignment between text pairs, and are lightweight in size and latency (e.g., ∼tens of millions of parameters in contrast to billions of parameters of performant decoder-only large language models).

**Masking Medium Words** Phrases such as "image of", "picture of", or "photo of" can appear in either alt-text or generated captions. We refer to such phrases as "medium phrases", as they describe the medium rather the contents of an image. We notice that the existence of such medium phrases adds noise to the sentence similarity score, as shown in Figure 3. A pair of sentences that each have a medium phrase are assigned a misleadingly high sentence similarity score by a sentence transformer, as they have been trained on a wide and diverse corpus of text, rather than on image captions. Hence, the existence of medium phrases may increase their attention to the topic of images or media, rather than the topic of the content of such images. Therefore, we neutralize the effect of medium phrases by removing them from both alt-text and generated captions. We express the operation of masking medium words in a text sample, $M(T)$, on text, $T$, as masking all possible contiguous subsequences of the text. where masking on a phrase, $t$, removes it if it is in the pre-determined list of medium phrases, $\mathcal{M} = \{$"image of", "picture of", "photo of", … $\}$, or keeps it otherwise.

Putting it all together, we define the SIEVE score function as:

$$f_{\text{SIEVE}}(I, T) = \max_{T_j^G \in G(I,r)} \langle S(M(T_j^G)), S(M(T)) \rangle \tag{8}$$

The dataset pruned using SIEVE with the top $k$ portion of its samples can be expressed as:

$$\mathcal{D}_{\text{SIEVE},k} = \text{prune}_{f_{\text{SIEVE}}}(\mathcal{D}, k) \tag{9}$$

We summarize our approach in Figure 2 and as psuedocode in Algorithm 1 in the Appendix.

Table 1: Zero-shot performance of CLIP models pretrained using various filtering strategies on *medium* and *large* scale pools of the DataComp benchmark. SIEVE fused with CLIPScore achieves competitive results on *medium* scale, and state-of-the-art performance on *large* scale.

| Scale | Filtering | Dataset Size | ImageNet | ImageNet dist. shifts | VTAB | Retrieval | Average over 38 datasets |
|---|---|---|---|---|---|---|---|
| Medium (128 Million) | No Filtering | 128M | 17.6 | 15.2 | 25.9 | 21.9 | 25.8 |
| | Basic Filtering | 30M | 22.6 | 19.3 | 28.4 | 25.1 | 28.5 |
| | LAION Filtering | 13M | 23.0 | 19.8 | 30.7 | 23.3 | 29.2 |
| | CLIPScore | 38M | 27.3 | 23.0 | 33.8 | 25.1 | 32.8 |
| | SIEVE | 24M | 29.4 | 25.0 | 35.2 | **28.9** | 34.6 |
| | SIEVE+CLIPScore | 24M | **30.3** | **25.4** | **36.2** | 27.8 | **35.4** |
| Large (1.28 Billion) | No Filtering | 1.28B | 45.9 | 37.8 | 42.6 | 41.9 | 43.7 |
| | Basic Filtering | 298M | 51.6 | 42.3 | 44.6 | 48.0 | 45.8 |
| | LAION Filtering | 130M | 55.3 | 45.3 | 51.0 | 49.5 | 50.1 |
| | CLIPScore | 384M | 57.8 | 47.4 | 53.8 | 46.6 | 52.9 |
| | SIEVE | 235M | 57.3 | 47.8 | 52.0 | **52.0** | 52.3 |
| | SIEVE+CLIPScore | 235M | **59.7** | **49.1** | **54.8** | 51.1 | **54.6** |

## 4 EXPERIMENTS

### 4.1 TRAINING AND EVALUATION

We utilize the DataComp benchmark to evaluate the utility of image-captioning models for multimodal dataset pruning. Two candidate pools are considered, the *medium* and the *large* scale, consisting of 128 million and 1.28 billion image-text pairs, respectively. To train CLIP models, we use DataComp's hyperparameters and architectures to standardize training (Gadre et al., 2023): $5 \times 10^{-4}$ learning rate, 500 iterations warmup, AdamW optimizer, for *medium* scale: ViT-B/32 image encoder (Dosovitskiy et al., 2021), batch size 4096, 128M training samples as a compute budget, and for *large* scale: ViT-B/16 image encoder, batch size 8192, 1.28B training samples as a compute budget. We evaluate the zero-shot performance on 38 downstream tasks, including classification and retrieval tasks (Radford et al., 2021; Kumar et al., 2022; Zhai et al., 2019).

For our captioning model, we utilize BLIP with ViT-B/16 image encoder pretrained on 14 million image-text pairs (Li et al., 2022), including Conceptual Captions (Sharma et al., 2018), Conceptual 12M (Changpinyo et al., 2021), SBU captions (Ordonez et al., 2011), COCO, and Visual Genome (Krishna et al., 2017). To compute the alignment between generated captions and alt-text, we use a lightweight distilled sentence transformer, all-MiniLM-L6-v2 (Wang et al., 2020), further finetuned using self-supervised contrastive learning on billions of text pairs.

### 4.2 MAIN RESULTS

Table 1 reports multiple baselines from DataComp (Gadre et al., 2023), including applying no filtering, basic filtering, and CLIPScore filtering. On the *medium* scale, SIEVE with an image-captioning model pretrained on 30 times less but curated data surpasses CLIPScore by 1.8% on average. We also fuse SIEVE with CLIPScore by applying min-max normalization to SIEVE alignment scores and CLIPScore, then taking the per-sample weighted average of both scores:

$$f_{\text{SIEVE+CLIP}}(I, T) = (1 - \alpha) \times \overline{f}_{\text{SIEVE}}(I, T) + \alpha \times \overline{f}_{\text{CLIP}}(I, T)$$

$$\text{s.t.} \qquad \overline{f}(I, T) = \frac{f(I, T) - \min_{(I_i, T_i) \in \mathcal{D}} f(I_i, T_i)}{\max_{(I_i, T_i) \in \mathcal{D}} f(I_i, T_i) - \min_{(I_i, T_i) \in \mathcal{D}} f(I_i, T_i)}$$

where the weight $\alpha$ used in the reported results is 0.5. Finally, we select the top 20% of samples. We observe that the fused approach improves performance on *medium* scale and achieves state-of-the-art performance on *large* scale. Moreover, SIEVE achieves the best performance on retrieval tasks on both *medium* and *large* scale experiments. In addition, while SIEVE without CLIPScore fusion surpasses CLIPScore on *medium* scale, this was not the case at *large* scale, demonstrating that 1) a method that performs well on a smaller scale might not do as well when the scale of the data and/or the model is increased, 2) CLIPScore from a pretrained model on a large, diverse but noisy dataset can also be useful to augment SIEVE for pruning, especially at larger scales.

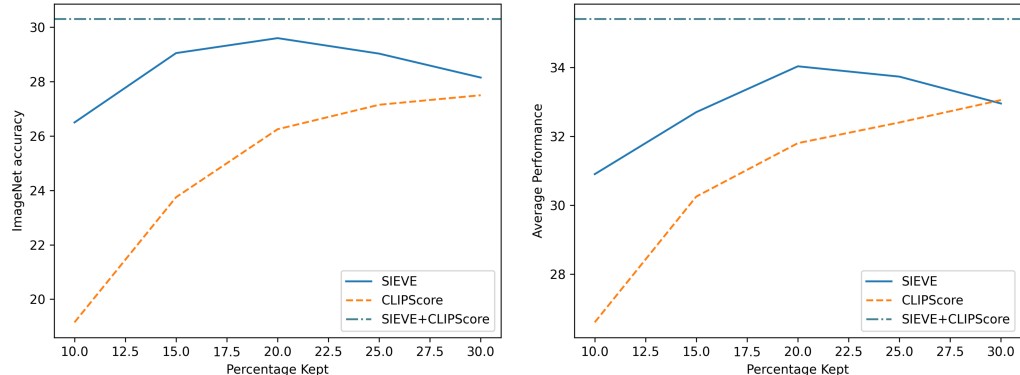

Figure 4: Evaluating CLIP models pretrained on different fractions of the top ranked samples based on our proposed approach (SIEVE), CLIPScore, and fusing SIEVE with CLIPscore (SIEVE+CLIPSCore), on *medium* scale.

Figure 5 shows the change in accuracy introduced by SIEVE as well as SIEVE+CLIPScore on each task compared to CLIPScore on *medium* scale, and Figure 7 of the Appendix shows for the *large* scale. We observe that in addition to outperforming on image retrieval tasks, Flickr (Young et al., 2014), and MS COCO (Lin et al., 2014), SIEVE's greatest performance boost comes from WingoGAViL (Bitton et al., 2022), a retrieval task which requires diverse reasoning skills, including general knowledge, common sense, and abstraction. This high performance can be attributed to SIEVE's preference towards keeping samples where the alt-text correctly describes visual concepts and their attributes and relations. SIEVE, especially when combined with CLIPScore, significantly outperforms on medical diagnosis tasks, Camelyon17 and PatchCamelyon. On *large* scale, SIEVE demonstrates a large boost on DollarStreet (Rojas et al., 2022), a dataset that shows pictures of household items from families of diverse ethnic and economic backgrounds.

SIEVE mainly underperforms in tasks requiring parsing text from images, such as MNIST (Lecun et al., 1998), SVHN (Netzer et al., 2011), and Rendered SST-2 (OpenAI, b), concluding that SIEVE is less likely to select image-text pairs that are useful for OCR tasks. In addition, SIEVE underperforms CLIPScore on context-based tasks like Country211 (OpenAI, a), a task assessing the geolocation capability of visual representations, demonstrating SIEVE's preference towards selecting samples based on the alignment of alt-text with visual concepts rather than context. Interestingly, when fusing with CLIPScore, we improve the performance of all these tasks while retaining the advantage of SIEVE especially at *large* scale (see Figure 7).

## 4.3 ABLATION STUDIES

We conduct studies on the *medium* scale pool and report the average of three runs per experiment.

**Pretraining data-distribution** We study the effect that the pretraining data distribution used to train the captioning model has on the quality of the alignment score. This is measured based on the downstream performance of the CLIP model trained using the selected image-text pairs. Two pretraining data distributions proposed by BLIP (Li et al., 2022) are investigated. The first uses 14 million curated image-text pairs, while the second uses an additional 115 million web images with noisy alt-text (Schuhmann et al., 2021). Although the original BLIP work reports higher captioning performance when pretraining on 115 million samples, our results in Table 2 indicate that for the purpose of dataset pruning, using curated image-text pairs results in a better alignment score than using a much larger noisy dataset. This highlights the importance of using a captioning model pretrained on higher quality data for large-scale dataset pruning.

**Text embedding space** In Table 3, we ablate over embeddings from different text models and show that embeddings from our selected sentence transformer perform better than embeddings from CLIP and BLIP text encoders. The CLIP text encoder was pretrained along with the CLIP vision encoder to map text and images to the same embedding space, and is used in diffusion models to condition image generation (Ramesh et al., 2022). However, we observe that the CLIP text encoder suffers

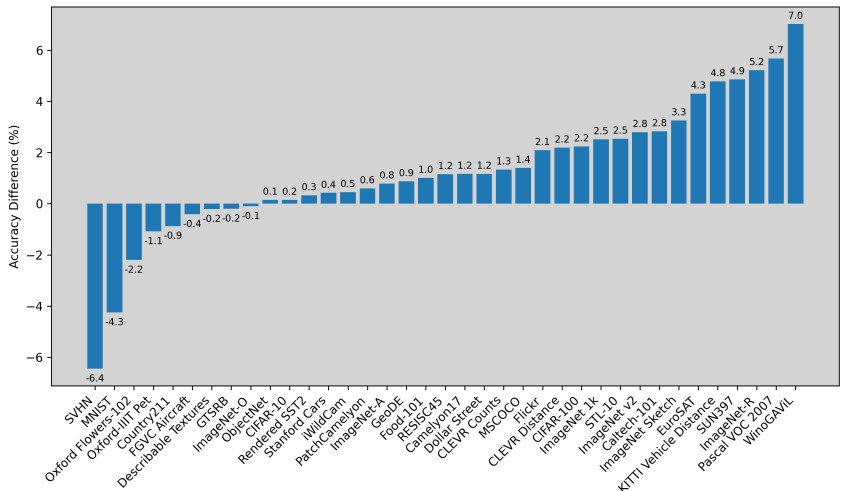

(a) SIEVE gain over CLIPScore on *medium* scale pool.

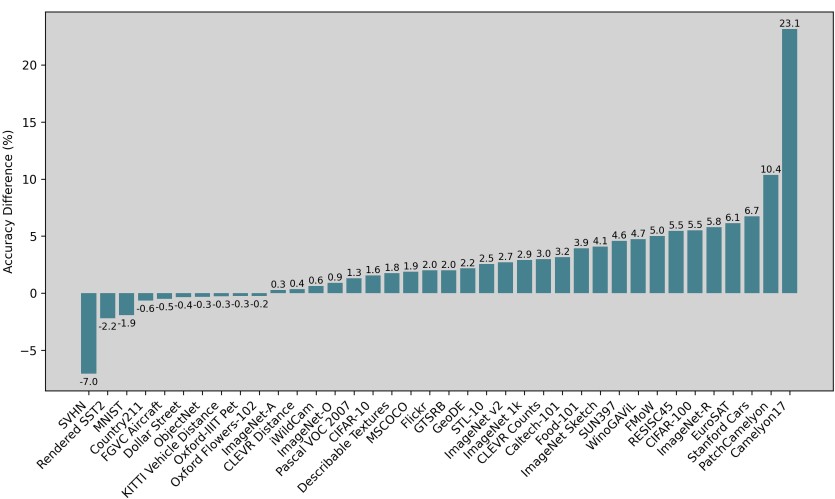

(b) SIEVE+CLIPScore gain over CLIPScore on *medium* scale pool.

Figure 5: The relative performance gain of SIEVE and SIEVE+CLIPScore relative to CLIPScore on 38 downstream tasks on the *medium* scale pool.

from poor semantic textual understanding, leading to a large drop in accuracy when used as a caption similarity measure. BLIP's text encoder performs better than that of CLIP, but the lightweight sentence transformer specifically pretrained on aligning semantically similar texts performs significantly better with $\geq 2\%$ improvements across various task types. In Figure 8 in the Appendix we show how cosine similarities of sentence similarity models result in better semantic textual clustering compared to CLIP and BLIP text encoders.

**Pruning percentage** We study the effect of the fraction of samples selected for pretraining. For each experiment, we compute the SIEVE alignment score and CLIPScore for each sample. The top-$k\%$ and pretraining CLIP models are then selected. Here, $k\%$ is set to 10%, 15%, 20%, 25% and 30%. Finally, we report the zero-shot performance on ImageNet and the average on 38 tasks in Figure 4. We observe that SIEVE achieves the best performance using 20% of the data, while CLIPScore peaks at 30% (similar to results reported in Gadre et al., 2023). Hence, pruning using SIEVE achieves better performance with less data, compared to CLIPScore.

**Number of generated captions and fusion with CLIPScore** We study the effect of using multiple captions per image to maximize the alignment of the generated captions with the alt-text. For nucleus sampling (Holtzman et al., 2020), we set the cumulative probability of the small-

Table 2: Effect of a caption generator's pretraining data-distribution on SIEVE. The 14M pretraining dataset consists of curated image-text pairs, while the 129M dataset includes an additional 115M noisy image-text pairs from LAION (Schuhmann et al., 2021).

| Percentage Kept | Caption Generator Pretraining Data | ImageNet | ImageNet dist. shift | VTAB | Retrieval | Average over 38 datasets |
|---|---|---|---|---|---|---|
| 10 | BLIP-129M | 23.00 | 20.60 | 30.20 | 21.40 | 30.00 |
| | BLIP-14M | **26.50** | **22.50** | **32.10** | **23.75** | **30.90** |
| 15 | BLIP-129M | 25.95 | 22.80 | 32.80 | 24.40 | 32.40 |
| | BLIP-14M | **29.05** | **24.60** | **33.35** | **26.95** | **32.70** |
| 20 | BLIP-129M | 27.85 | 23.65 | 33.45 | 26.35 | 33.05 |
| | BLIP-14M | **29.60** | **24.93** | **35.07** | **28.57** | **34.03** |

Table 3: Effect of the sentence encoder on the performance of SIEVE. CLIP uses the text encoder from ViT-L pretrained on 400M samples (Schuhmann et al., 2021), BLIP uses the text encoder pretrained on the curated 14M samples defined in Li et al. (2022), and Sentence Transformer uses a language model pretrained on billions of sentences (Wang et al., 2020). Each encoder encodes both the generated captions and the alt-text where the textual semantic alignment is computed.

| Text Encoder | ImageNet | ImageNet dist shift | VTAB | Retrieval | Average over 38 datasets |
|---|---|---|---|---|---|
| CLIP | 18.00 | 15.65 | 26.90 | 20.95 | 25.90 |
| BLIP | 27.20 | 22.10 | 32.70 | 25.45 | 31.85 |
| Sentence Transformer | **29.60** | **24.93** | **35.07** | **28.57** | **34.03** |

est set of words to 0.9, and the minimum and maximum sequence lengths to 5 and 20, respectively. We study the effect of sampling 1, 2, 4, and 8 captions. For each input image-text pair, we assign the maximum alignment score between the alt-text and the generated captions. We observe in Table 6 that increasing the number of generated captions improves the performance on downstream tasks. We reason that due to the many-to-many relationship between images and captions, generating more captions increases the probability of matching a hard aligned alt-text. We also investigate the effect of fusing the SIEVE alignment score with CLIP-Score in Table 6. Each score is independently normalized, and a weighted average is applied between the two scores. Finally, the top 20% of samples ranked by SIEVE+CLIPScore are selected. We observe that a weight of 0.5 achieves the best performance on 38 downstream tasks.

## 5 CONCLUSION

We introduce a novel method, SIEVE, that enables pruning large-scale noisy web-crawled image-text datasets. We propose utilizing synthetic captions from image-captioning models pretrained on small, diverse, and curated datasets to evaluate the alignment of noisy image-text pairs. Using the embedding space of a lightweight sentence transformer, we compute an alignment score between generated captions and alt-text. We demonstrate the utility of SIEVE by achieving state-of-the-art performance on the *large* scale setup of the Data-

Figure 6: Effect of the number of generated captions and the weight of CLIPScore on zero-shot performance of pretrained CLIP models, on *medium* scale.

| Generated Captions | CLIP weight | ImageNet | Average over 38 datasets |
|---|---|---|---|
| 1 | 0.0 | 28.60 | 32.50 |
| 2 | 0.0 | 29.00 | 33.40 |
| 4 | 0.0 | 29.53 | 33.70 |
| 8 | 0.0 | **29.60** | **34.03** |
| 8 | 0.3 | 30.10 | 34.40 |
| | 0.5 | **30.35** | **35.15** |
| | 0.7 | 30.25 | 34.35 |

Comp benchmark. In the future, we would like to investigate the effect of applying text spotting Radenovic et al. (2023); Maini et al. (2023) on the image before generating synthetic captions to reduce the number of image-text pairs where the image contains text that has high intersection with the alt-text.

REPRODUCIBILITY

We have provided hyperparameter details to prune the dataset and train the CLIP model in Section 4.

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

## A    APPENDIX

---

**Algorithm 1** SIEVE Pseudocode to filter image-text datasets.

---

**Require:** Dataset $\mathcal{D}$, Fraction $k$ fraction to prune, Caption generator $G$, Sentence transformer $S$, Set of medium phrases $\mathcal{M}$, Number of captions to generate $r$
**Ensure:** Pruned dataset $\mathcal{D}_{\text{SIEVE}}$
 1: **for** each $I, T$ in $\mathcal{D}$ **do**                                      ▷ Loop over image-text pairs in the dataset
 2:     $e \leftarrow S(T)$                                                  ▷ Obtain sentence embedding of text label
 3:     $\mathcal{V} \leftarrow \{\}$                                               ▷ Initialize empty set of scores
 4:     $\mathcal{T}^G \leftarrow G(I, r)$                                        ▷ Generate a set, $\mathcal{T}^G$, of $r$ captions
 5:     **for** each $T^G$ in $\mathcal{T}^G$ **do**                             ▷ Loop over generated captions
 6:         $T^{G'} \leftarrow M(T^G)$                                          ▷ Mask medium words
 7:         $e^G \leftarrow S(T^{G'})$                          ▷ Obtain sentence embeddings of generated caption
 8:         $v \leftarrow \langle e^G, e \rangle$                               ▷ Compute sentence similarity
 9:         $\mathcal{V} += v$                                                 ▷ Append to set of scores
10:     **end for**
11:     $f_{\text{SIEVE}}(I, T) \leftarrow \max(\mathcal{V})$                     ▷ Obtain maximum score
12: **end for**
13: Rank $\mathcal{D}$ by $f_{\text{SIEVE}}(I, T)$ in descending order to obtain $\texttt{rank}_{\text{SIEVE}}(x)$
14: $\mathcal{D}_{\text{SIEVE}} \leftarrow$ top $k\%$ of $\mathcal{D}$ based on $\texttt{rank}_{\text{SIEVE}}(x)$
15: **return** $\mathcal{D}_{\text{SIEVE}}$

---

Table 4: Intersection of ImageNet-based filtering and text spotting with SIEVE on *medium* scale. For intersecting with ImageNet samples, we use the same sampling approach proposed in Data-Comp Gadre et al. (2023). For intersecting with texting spotting, 1) we first utilize a text detector, to detect and compute the percentage of image covered by text, 2) Next, we rank samples in ascending order of the percentage of pixels covered by text, 3) finally, we keep the top 80% of the samples and intersect either with top 30% CLIPScore or top 20% top SIEVE samples.

| Filter | ImageNet | ImageNet dist. shift | VTAB | Retrieval | Average over 38 datasets |
|---|---|---|---|---|---|
| CLIPScore | 27.30 | 23.00 | 33.80 | 25.10 | 32.80 |
| SIEVE | **29.60** | **24.93** | **35.07** | **28.57** | **34.03** |
| CLIPScore ∩ ImageNet | **29.70** | 23.90 | **34.60** | 23.10 | **32.80** |
| SIEVE ∩ ImageNet | 28.95 | 23.30 | 33.05 | **26.60** | 32.05 |
| CLIPScore ∩ Text Spotting | 29.75 | 24.10 | **35.65** | 24.95 | **34.05** |
| SIEVE ∩ Text Spotting | **30.10** | **25.05** | 34.15 | **28.35** | 33.90 |

Table 5: Intersection of ImageNet-based filtering on *large* scale. We achieve state-of-the-art performance on ImageNet, ImageNet out-of-distribution and retrieval tasks.

| Filter | ImageNet | ImageNet dist. shift | VTAB | Retrieval | Average over 38 datasets |
|---|---|---|---|---|---|
| CLIPScore | 57.8 | 47.4 | 53.8 | 46.6 | 52.9 |
| SIEVE | 57.3 | 47.8 | 52.0 | **52.0** | 52.3 |
| SIEVE+CLIPScore | **59.7** | **49.1** | **54.8** | 51.1 | **54.6** |
| CLIPScore ∩ ImageNet | 63.1 | 50.8 | **54.6** | 49.8 | **53.7** |
| SIEVE ∩ ImageNet | 61.2 | 49.2 | 51.3 | 51.4 | 51.4 |
| SIEVE+CLIPScore ∩ ImageNet | **63.8** | **51.4** | 53.1 | **53.3** | 53.6 |

Table 6: Intersection-Over-Union between the unique ids of top-$k$ samples selected by CLIPScore and 1) SIEVE, 2) SIEVE+CLIPScore on *medium* scale

| Top-$k$% | SIEVE | SIEVE + CLIPScore |
|---|---|---|
| 10 | 16.00% | 27.18% |
| 20 | 29.99% | 44.00% |
| 30 | 39.54% | 56.93% |

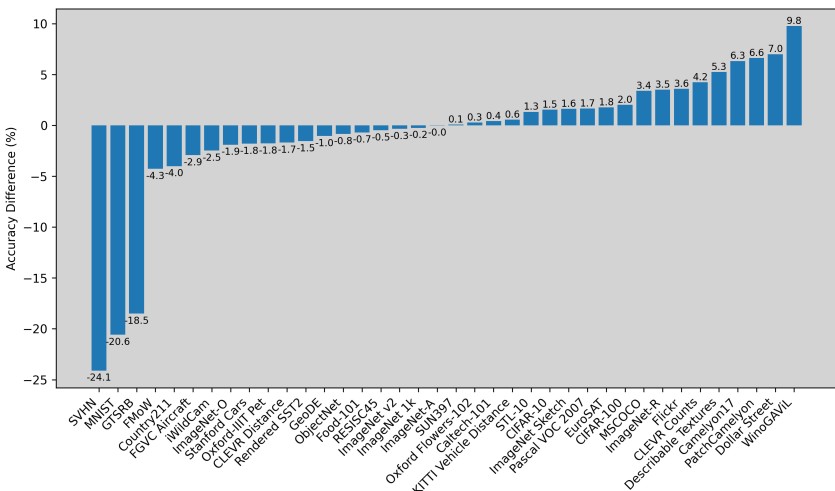

(a) SIEVE gain over CLIPScore on *large* scale pool.

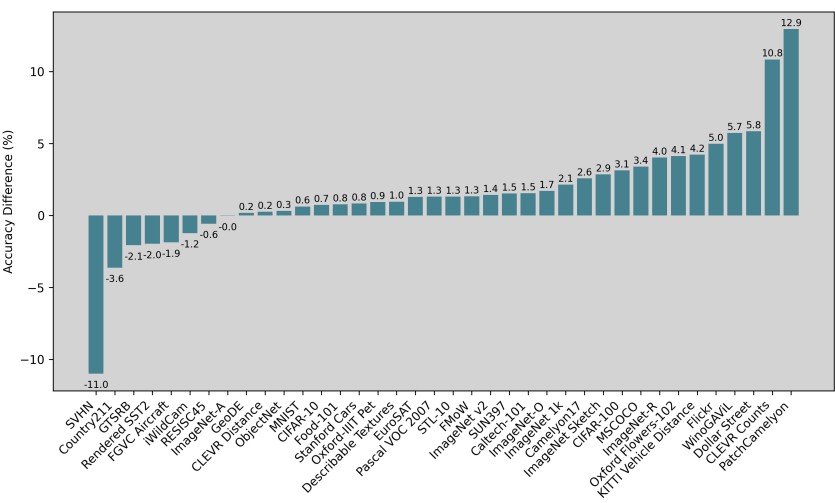

(b) SIEVE+CLIPScore gain over CLIPScore on *large* scale pool.

Figure 7: The relative performance gain of SIEVE and SIEVE+CLIPScore relative to CLIPScore on 38 downstream tasks on "large" scale pool.

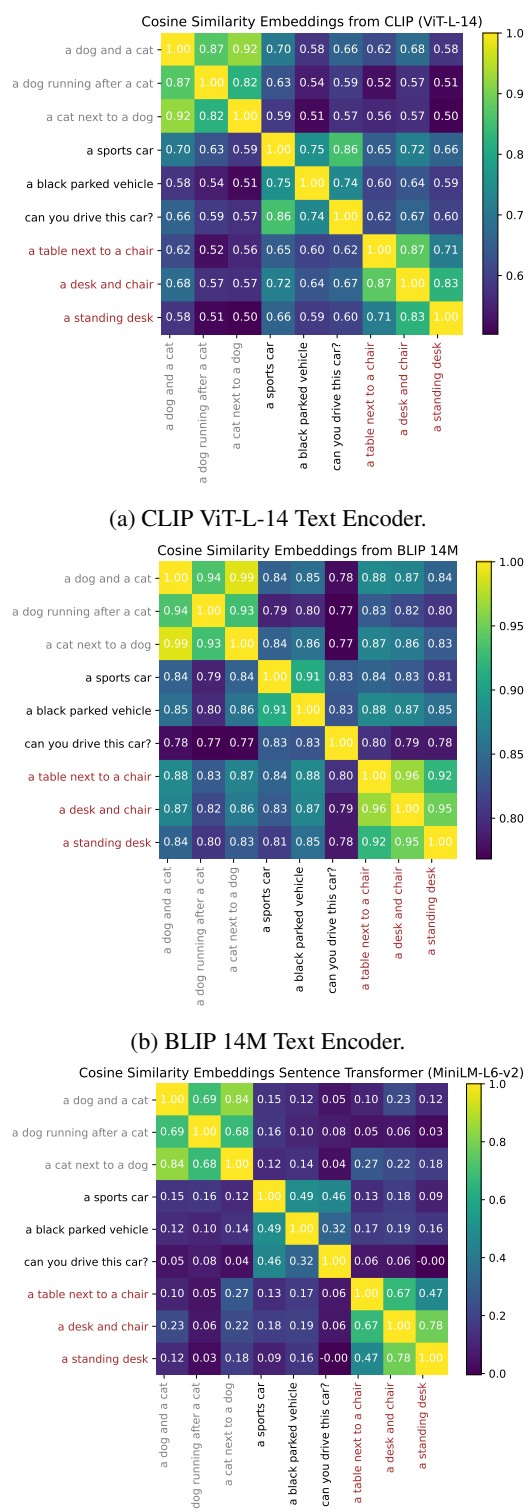

(a) CLIP ViT-L-14 Text Encoder.

(b) BLIP 14M Text Encoder.

(c) Sentence Transformer all-MiniLM-L6-v2.

Figure 8: Confusion matrix of cosine similarity illustrating the performance of each text encoder in similarity. We show 9 sentences split into 3 groups of consecutive sentences, where sentences within each group describe similar concepts. We observe that Sentence Transformer has better semantic textual clustering compared to CLIP and BLIP text encoders.

