# OpenReview forum: "SIEVE: Multimodal Dataset Pruning using Image-Captioning Models"
_ICLR.cc/2024/Conference — ICLR 2024 Conference Withdrawn Submission_

### Official Review · Reviewer_cmps · 2023-10-22

**Soundness:** 2 fair
**Presentation:** 3 good
**Contribution:** 1 poor
**Rating:** 3
**Confidence:** 5

**Summary:**

The paper proposes a data filtering method using an image captioning model. First, it trains an image captioning model on a small and high-quality dataset. Second, it uses the model to generate captions for the images to be filtered. Third, it computes the similarities between the generated captions and the images' original alt-texts, and filters out the low-score image-text pairs.

**Strengths:**

It is an interesting attempt to go beyond CLIPScore based filtering and explore other score based filtering methods.

**Weaknesses:**

1. The proposed method is not well motivated. The authors argue CLIPScore has three limitations, and propose a caption-text score, where caption is generated by a captioning model on an image, and text is alt-text. However, the caption-text score could also have similar limitations - the proposed method could also miss hard samples and the captioning model could also be biased, which are not discussed in the paper.
2. Even if the hypotheses in "Section 3 - Image-Captioning" are true, they do not necessarily result in a better filtering metric than CLIPScore. We can hypothesize the similar statements for CLIPScore. The large-scale results in Table 1 supports this - SIEVE's avg performance 52.3 is worse than CLIPScore's avg performance 52.9.
3. As mentioned in the paper (Sec 4.2), the results between medium and large scales are not consistent. The reviewer suspects result variance could partially explain that. How many models are trained for each filtering setting to get the quantitative results, and what is the variance?

**Questions:**

See weaknesses.

---

> ### Author Response · Authors · 2023-11-17
> **Author's reply to Reviewer cmps**
>
> We thank the reviewer for their feedback and would like to address their concerns:
>
> **"The authors argue CLIPScore has three limitations, and propose a caption-text score, where caption is generated by a captioning model on an image, and text is alt-text. However, the caption-text score could also have similar limitations"**
>
> 1. We agree with the reviewer that SIEVE’s pruning signal can still result in missed hard samples and we will add a section discussing the bias of SIEVE. Nevertheless, we demonstrate that SIEVE and CLIP Score’s pruning signals are highly complementary:
>     1. Compared to CLIP Score, SIEVE is more likely to focuse on image content rather than context as shown in Figure 1 of the paper. This is clear in the per-task performance on both medium and large scale where SIEVE on its own surpasses CLIP score on retrieval tasks by a large margin (+3.8%, 5.4%), while CLIP Score due to its large pretraining data, performs better on context-based tasks like identifying the geographical location from an image.
>      2. We show that SIEVE when augmented to CLIPScore, beats CLIPScore on the main tasks and on Average on both medium and large scales. Hence, SIEVE is a useful filtering signal.
> 2. We also demonstrate in our ablation studies (table 3) that compared to CLIP and BLIP text encoders, our sentence similarity embedding space can bridge the diversity gap between the alt-text and the captioner output, improving our chances of detecting well-aligned hard samples.
>
> **"The large-scale results in Table 1 supports this - SIEVE's avg performance 52.3 is worse than CLIPScore's avg performance 52.9"**
> 1. Our goal is not to beat CLIP Score but to provide a complementary ranking signal focused on image content aiming to reduce false positives and negatives. Fusing CLIPScore with SIEVE improves overall performance by 2.6% and 1.7% on medium and large scale respectively.
> 2. The reported average performance is conducted on 38 downstream tasks concealing per-task performance. For instance, SIEVE on its own performs significantly better on retrieval tasks and this performance gap increases from medium (+3.8%) to large scale (5.4%).
> 3. We argue that even though SIEVE is pre-trained only on 14 million image-text pairs, its performance compared to CLIPScore (pretrained on 400 million image-text pairs) is within less than 1% on the large scale, which is very promising.
>
> **"As mentioned in the paper (Sec 4.2), the results between medium and large scales are not consistent. The reviewer suspects result variance could partially explain that. How many models are trained for each filtering setting to get the quantitative results, and what is the variance?"**
>
> 1. For all our ablation studies, we run 3 experiments and report the average performance in the paper. The variance of SIEVE is 0.29 on medium scale  and 0.31 on large scale. For the submission on the leaderboard, we are only able to submit one pretrained model (seed=1).
> 2. The scale of experiments in terms of the candidate pool size, model capacity, and compute budget (maximum number of samples seen) can have a significant effect on the ranking of pruning methods. Due to the biases of each ranking method, a method that performs well on a small or medium scale does not necessarily perform better on a larger scale. This is shown both in the DataComp paper where, for instance, image-based filtering performs better than LAION filtering on medium scale but the order is reversed on large scale.
> 3. With respect to average performance, the order of CLIPScore and SIEVE are reversed between medium and large but the gap is within less than 1%. Nevertheless, the performance of SIEVE on retrieval tasks consistently improves from 3.8% to 5.4% from medium to large scale. Finally, fusing CLIPScore with SIEVE leads to a 2.6.7% improvement on medium and 1.7% on large over CLIPScore.

---

### Official Review · Reviewer_pF34 · 2023-10-27

**Soundness:** 3 good
**Presentation:** 3 good
**Contribution:** 3 good
**Rating:** 8
**Confidence:** 4

**Summary:**

This paper studies dataset curation in the context of training CLIP moddels (which contrast images and text). Such models are typically trained on large datasets containing pairs of images and captions collected from the web, and curating such datasets has been an open research problem. The authors propose a new method for curating large scale image-text datasets, called SIEVE. The method works as follows. First, the authors use an image captioning model to generate captions for the images in the dataset. Next, the generated captions are compared to the original captions using a sentence similarity model. This score, optionally combined with other techniques (e.g. CLIPScore, the alignment between the image and original caption according to a trained CLIP model), shows to be a useful signal for selecting samples from a dataset. The authors present several experiments on DataComp, a dataset filtering benchmark, and achieve state of the art performance on both the medium and large scales of this competition.

**Strengths:**

This paper has several strengths.

1. Firstly, the results are quite strong, providing substantial gains over strong data curation baselines. The authors obtain state of the art on a the challenging DataComp benchmark.
2. The proposed method is quite simple and easy to implement. It is also not very computationally expensive.
3. Both CLIP models and better methods for designing datasets are vibrant research directions, and this paper present solid advances in both. Therefore I believe it would be of interest to many in the community

**Weaknesses:**

1. It is unclear how scalable the method is. The fact that the improvements seen at medium scale are larger than the improvements seen at large scale of DataComp are somewhat concerning for the scalability of the method. The paper would be stronger if these gains were consistent or grew as scale increased.
2. While the authors present several ablations, I believe the captioning model is a central part of their pipeline. As such, I think the paper would be stronger if the authors explored more diverse captioning models.

**Questions:**

Two of the main downsides listed by the authors for using CLIPScore models (false positives and false negatives) seem to be mitigated by better CLIP models. In the future where we have better CLIP models, do you expect CLIP models to be enough? Also, if these were the main causes of the shortcomings of using CLIPScore, shouldn't we expect that using better CLIP models would lead to better filtering? This is in contrast to experiments in previous literature.

---

> ### Author Response · Authors · 2023-11-17
> **Author's reply to Reviewer pF34**
>
> We thank the reviewer for the insightful comments and constructive criticism. We would like to address their concerns below:
>
> **"It is unclear how scalable the method is. The fact that the improvements seen at medium scale are larger than the improvements seen at large scale of DataComp are somewhat concerning for the scalability of the method. The paper would be stronger if these gains were consistent or grew as scale increased."**
>
> While the standalone performance of SIEVE is better on medium scale compared to large scale (relative to CLIPScore), the average performance for instance hides the fact that SIEVE does significantly better on retrieval tasks at scale compared to CLIPScore (3.8% on medium versus 5.4% on large). In addition, since our goal is not to replace CLIPScore but to provide a complementary ranking signal focused on image content rather than context, we demonstrate that fusing CLIPScore with SIEVE improves performances compared to CLIPScore by 2.6% on medium, while the improvement on large is at 1.7%
>
> **"While the authors present several ablations, I believe the captioning model is a central part of their pipeline. As such, I think the paper would be stronger if the authors explored more diverse captioning models."**
>
> We agree with the reviewer that exploring multiple captioning models would strengthen the paper. Since BLIP provides models pretraining on 14M curated set and 129M uncurated set, we thought it gives us a good opportunity to investigate the captioner pre-training data-distribution on SIEVE. We are currently conducting experiments using other captioning models.

---

### Official Review · Reviewer_sFAh · 2023-10-31

**Soundness:** 2 fair
**Presentation:** 3 good
**Contribution:** 2 fair
**Rating:** 3
**Confidence:** 4

**Summary:**

The paper proposes a method for pruning large text-image datasets for pretraining models. Given a dataset with uncurated text-image samples, the proposed method consists of generating multiple captions from an image, matching the generated captions with the original associated text, and discarding samples with low text cosine similarity score. The method is evaluated on the DataComp benchmark on the “medium” and “large” settings.

**Strengths:**

The proposed idea is simple and seems to remove miss-aligned text-image samples well enough to increase the performance of the downstream tasks with respect to a set of baselines.

**Weaknesses:**

1. First, the proposed model was submitted as part of the DataComp competition, where other proposed models from other teams were submitted as well. On the medium-scale experiment, SIEVE is currently ranked in the 6th position (5th by the ICLR submission time). However, the paper fails to acknowledge the other competitor models and does not report their results. This lack of transparency regarding how the model performs with respect to other submissions (and only reporting results on the baselines) is concerning and provides a partial view of the state of the art to potential readers.

2. I wonder how much the pruned model can be claimed to be trained on a 24 million dataset while, at the same time, the captioner model used to prune the model has been trained on 14 million image-text pairs and the text matching network on billions of text pairs. Clearly, the final model is leveraging the information from the captioner and text matching network, so I don’t see a clear justification to not count the data used on those two subcomponents of the proposed method as training data. I think a clear and honest discussion about the data used for the whole process is necessary.

3. The paper claims repeatedly that the dataset used for training the captioner model is “curated”, “higher data quality”, or “well-aligned”. However, the datasets used for training the captioner are GCC3M, GCC12M, SBU, COCO, and Visual Genome. From the total of 14 million images, about 12 million may come from GCC3M or GCC12M. Both of these datasets are neither curated nor well-aligned and they suffer from the same problems as LAION. In fact, the collection processes between GCC and LAION are very similar.

**Questions:**

It would be interesting to evaluate how bias (e.g. gender bias) behaves with the pruned dataset with respect to the original dataset: i.e. can bias be reduced by pruning, or is bias increased/unchanged?

---

> ### Author Response · Authors · 2023-11-17
> **Author's reply to Reviewer sFAh**
>
> We thank the reviewer for their feedback and would like to address their concerns below
>
> **"the proposed model was submitted as part of the DataComp competition, where other proposed models from other teams were submitted as well. The paper fails to acknowledge the other competitor models and does not report their results."**
>
> 1. At the time of submission, T-MARS[1], an orthogonal concurrent pruning approach which relies on text masking before applying CLIPScore, was the only medium-scale submissions that was available on arxiv. We cite this approach in our literature review. We also mention in our future works section that we aim to utilize text masking in SIEVE before captioning to improve SIEVE. Finally, we conduct additional experiments showing the effect of text masking on SIEVE and report the results in Table 4 in the appendix. Works that we did not cite did not have publications associated with them at the time of submission.
>
> 2. All other methods either had no associated publication or was arvix a day before the conference deadline:
> “The Devil is in The Details” (DID) paper only appeared on arxiv on September 27th, one day before ICLR submission deadline and we only noticed it a couple of weeks after the deadline: https://arxiv.org/abs/2309.15954
> “Data Filtering Networks” paper only appeared on arxiv on Septembet 29th, one day after ICLR submission deadline: https://arxiv.org/abs/2309.17425
>
> 3. The novelty of SIEVE is that it enables pruning image-text datasets with captioning models pretrained on small but curated datasets.  “The Devil in the Details” (DID) submission focuses on fusing many ranking signals to improve accuracy, a phenomenon that was shown in the DataComp paper especially at scale. We demonstrate that SIEVE provides a promising ranking signal and demonstrate that it is indeed complementary to CLIP Score at medium and large scale. Moreover, DID aligns CLIP's pretraining data distribution with the DataComp evaluation set (downstream tasks). We argued in the paper that selecting samples that match the distribution of downstream tasks encourages overfitting to the evaluation set and, thus limits generalization to other downstream tasks. Without that distribution alignment, our approach beats DID by a large margin on average performance (32.8% versus 35.4%).
>
> 4. The Data Filtering Networks (DFN) submission, uses a CLIP model pre-trained  on a non-public 357 million human curated dataset (as the paper stated that the labels were “human verified”), to rank the DataComp samples. Compared to DFN, our captioner is pretrained on 25x less data but still holds second place on large-scale. Moreover, DFN finetunes on the training set of ImageNet, the main evaluation task of DataComp. Average performance of DFN without fine-tuning on ImageNet is within 0.5% of our approach (54.6% versus 55.0%).
>
> 5. With the exception of DFN, all other methods do not report results at the large scale, while our paper shows results for both medium and large scales. The change of ordering of pruning methods is shown in the original Datacomp paper where, for example, image-based filtering performs better than LAION filtering on medium scale but worse on large-scale.
>
>
> **"I wonder how much the pruned model can be claimed to be trained on a 24 million dataset while, at the same time, the captioner model used to prune the model has been trained on 14 million image-text pairs and the text matching network on billions of text pairs."**
>
> Our claim is that we pretrain on 14 million image-text pairs, not 24 million. Compared to text data, curating image-text pairs is very expensive as it involves aligning two modalities. Our captioner is pre-trained on 14 Million pairs, where 12 Million samples are from CC12M where the curation process involves uni-modal and multimodal curation. On the other hand, the pretraining task for the sentence similarity model is simply to determine whether two sentences originate from the same document. Constructing such unlabelled text dataset for pretraining the sentence similarity model is trivial and does not require multimodal alignment. We use a publicly available sentence similarity model consisting of only 22 million parameters.

---

> > ### Author Response · Authors · 2023-11-17
> > **Author's reply to Reviewer sFAh**
> >
> > **"The paper claims repeatedly that the dataset used for training the captioner model is “curated”, “higher data quality”, or “well-aligned”. However, the datasets used for training the captioner are GCC3M, GCC12M, SBU, COCO, and Visual Genome. From the total of 14 million images, about 12 million may come from GCC3M or GCC12M. Both of these datasets are neither curated nor well-aligned and they suffer from the same problems as LAION. In fact, the collection processes between GCC and LAION are very similar."**
> >
> > We highly disagree with the reviewer that GCC and LAION filtering are very similar. GCC is highly curated compared to LAION dataset
> >
> > **LAION curation involves the following steps [1]**:
> > 1. Unimodal: Removing samples with less than 5 character alt-text length or less than 5KB image size along with duplicate removal
> > 2. Multimodal: Use CLIPScore to exclude samples with low clip score
> >
> > **While, GCC curation involves the following steps [2,3]**:
> > 1. Unimodal: Image-based filtering discards images based on encoding format, size, and aspect ratio. It only keeps JPEG images where both dimensions are greater than 400 pixels, and the ratio of the larger to smaller dimension is no more than 2.
> > 2. Unimodal: Text-based filtering
> >     1. Candidates with no determiner, no noun, or no preposition are discarded; candidates with a high noun ratio are also discarded;
> >      2. Use a vocabulary of 1B token types, appearing at least 5 times in the English Wikipedia, and discard candidates that contain tokens that are not found in this vocabulary
> >      3. Predefined boiler-plate prefix/suffix sequences matching the text are cropped, e.g. “click to enlarge picture”, “stock photo”;  also drop text which begins/ends in certain patterns, e.g. “embedded image permalink”, “profile photo”.
> >      4. This filter discards around 97% of the incoming candidates.
> > 3. Multimodal: Image-text filtering:
> >     1. Filters out candidates for which none of the text tokens can be mapped to the content of the image: Image classification models available via the Google Cloud Vision APIs are used to assign multiple class labels to each image, using an image classifier with a large number of labels (order of magnitude of 10^5 ). Images are generally assigned between 5 to 20 labels
> >     2. Those generated labels are then matched against the candidate text, taking into account morphology-based stemming as provided by the text annotation.
> >     3. If there is no overlap between the captions and the labels the image-text sample is discarded.
> >     4. This filter discards around 60% of the incoming candidates.
> >
> > The main difference between GCC3M[2] and GCC12M[3] is that the unimodal filters are relaxed but the multimodal filters are kept the same.
> >
> > Our ablations in Table 2 clearly indicate that the pretrained captioner of SIEVE with smaller but curated image-text pairs (14M) provides significantly better pruning signal compared to a captioner pre-trained with large noisy image-text pairs (129M).
> >
> > **References:**
> > 1. Schuhmann et al. "LAION-400M: Open Dataset of CLIP-Filtered 400 Million Image-Text Pairs"
> > 2. Sharma et al. "Conceptual Captions: A Cleaned, Hypernymed, Image Alt-text Dataset For Automatic Image Captioning"
> > 3. Changpinyo et al. "Conceptual 12M: Pushing Web-Scale Image-Text Pre-Training To Recognize Long-Tail Visual Concepts"